# The Phosphoproteomic Response of Pepper (*Capsicum annuum* L.) Seedlings to Selenium Stress

**Jiahua Wu [†], Shixian Guo [†], Jing Wang, Jiaojun He, Xingfu Li and Yihua Zhan ***

The Key Laboratory for Quality Improvement of Agricultural Products of Zhejiang Province, College of Advanced Agricultural Sciences, Zhejiang A & F University, Hangzhou 311300, China; 19550176782@163.com (J.W.); sxguo1115@163.com (S.G.); m15824467544@163.com (J.W.); hejiaojun@stu.zafu.edu.cn (J.H.); 17388583621@163.com (X.L.)

* Correspondence: yhzhan@zafu.edu.cn
[†] These authors contributed equally to this work.

**Abstract:** Excessive selenium has gradually become a potential environmentally hazardous element for all organisms. Limited knowledge is available regarding the toxic mechanism of selenium in pepper, so the quantitative proteomics of phosphorylation was studied by Tandem Mass Tag approaches. A total of 4434 phosphorylation sites were identified on 2058 proteins, of which 3749 sites of 1919 proteins contained quantitative information. In the Se/mock (seedlings without Se treatment) comparison group, the number of upregulated phosphoproteins (658) was significantly higher than that of the downregulated ones (61). Systematic bioinformatics analysis, including protein annotation, functional classification, subcellular localization, and cluster analysis was performed. A total of 33 over-represented motifs were found in serine phosphorylation, and the most frequent motif was 'sP' (308 occurrences). According to KEGG enrichment analysis, the upregulated phosphorylated proteins (DPPs) were most strongly associated with the 'phenylpropanoid biosynthesis' and 'nicotinate and nicotinamide metabolism' pathways, while those that were downregulated were associated with the 'ABC transporters' and 'plant hormone signal transduction' pathways. Our data can provide new insights for evaluating the response mechanism of plants to selenium pollution and improving their resistance to selenium.

**Keywords:** pepper; phosphorylation; Se treatment; differentially phosphorylated protein; phenylpropanoid biosynthesis

## 1. Introduction

Selenium is a rare element essential for animals and humans, as well as being a nutritional element in plant growth [1]. A deficiency of selenium can cause a variety of serious health problems, and moderate amounts of selenium can prevent diseases, delay aging, and resist damage caused by various external stresses on the body [2]. Selenium in animals and humans mainly comes directly or indirectly from plants, so plants are known as biochemical factories for the synthesis of natural organic selenium compounds, playing a crucial part in the ecological chain of selenium conversion in nature [3,4]. Therefore, studying the molecular mechanism of selenium signaling in plants can promote the development of selenium rich foods and provide a theoretical basis for the absorption of plant-based selenium rich foods by the human body. Plants mainly absorb water-soluble selenium from soil solutions, including selenate, selenite, and organic selenium, which is mainly composed of selenate ($SeO_4^{2-}$) or selenite ($SeO_3^{2-}$) [5,6]. Moderate Se is beneficial to organisms, especially to improve resistance to heavy metal stress. For example, wheat selenium-binding protein TaSBP-A alleviated the oxidative stress by Cd [7]. Selenium increased the resistance of human kidney cells to arsenic stress by modifying phosphorylated proteins and reactive oxygen species [8]. In *Chlamydomonas reinhardtii*, selenium alleviated chromium-induced toxicity [9]. However, excessive selenium can lead to selenium poisoning in organisms [10].

Research has found that 8–16 mg/L selenium treatment can significantly inhibit barley germination [11], while 4–6 mg/L selenium can significantly inhibit the growth of soybean seedling roots and buds [12].

The increasing use of selenium in human activities, such as fertilizer application, mining, and metallurgy, has brought a large amount of selenium into soil and surface water, resulting in a rapid increase in surface selenium content [4,10]. At present, there are relatively few reports of Se toxicity in plants, and preliminary studies have found that the toxic effects of Se on plants are to some extent similar to those of heavy metals, causing growth inhibition and decreased metabolism in plants [13,14]. Due to the similarity in chemical properties between Se and S, Se competes with S and is transported into plants through sulfate transport proteins on the root plasma membrane [15]. Subsequently, part of $SeO_4^{2-}$ is reduced to $SeO_3^{2-}$ and converted to organic Se, which is further converted to selenide and volatilized. The other part of $SeO_4^{2-}$ metabolizes into selenocysteine (SeCys) or selenomethionine (SeMet) through the sulfur assimilation pathway in the chloroplast [16]. Toxicity is produced through two mechanisms: one is deformed selenoprotein [17,18], and the other is oxidative stress [11,16,19]. Most studies on selenium toxicity mainly analyze its toxicity mechanism from the perspective of plant oxidative damage caused by selenium, but further research on the signal transduction of selenium stress and the molecular mechanism of plant response to stress is still limited.

Proteomic analysis can provide extensive information on protein types and reveal the dynamic systems of overall biological changes [20]. As a branch of proteomics, phosphorylated proteomics is widely used in the study of plant organ/tissue development and stress-response accumulation [21,22]. Phosphorylation is an important form of post-translational modification of proteins, and phosphorylation of threonine, serine, and tyrosine residues has been widely demonstrated [23]. For instance, *Arabidopsis thaliana* photophosphorylpyruvate carboxylase (PEPCK) was phosphorylated at Ser55, Thr58, and Thr59 under light induction [24].

Pepper (*Capsicum annuum* L.), an annual herb of the Solanaceae family, is rich in minerals (calcium) and vitamins [25]. The large accumulation of carotenoids, capsaicins, and flavonoids in pepper determines the enormous value of chili fruits in natural colorants, cosmetics, and medical supplies [26]. Liu et al. performed a phosphoproteomic analysis of its fruit development and analyzed the dynamic changes of phosphorylated proteins in signaling transduction pathways [27]. However, the role of pepper protein phosphorylation in signal transduction under selenium stress is still obscure. In this study, to explore the toxic mechanism of selenium in pepper, a comprehensive phosphorylation proteomic analysis in pepper in response to selenium stress was conducted by Tandem Mass Tag (TMT) labeling approaches. Our data will provide candidate phosphorylated proteins related to selenium stress resistance.

## 2. Materials and Methods

### 2.1. Plant Materials and Treatments

The plant material used in study was *C. annuum* 8 #, provided by Fujian Agriculture and Forestry University. After 30 min of 1% sodium hypochlorite sterilization, seeds were sown in sterilized soil. Plants were grown in the Pingshan greenhouse of Zhejiang A & F University with 12 h of light (150 m$^{-2}$ s$^{-1}$) at 26 °C and 12 h of darkness at 23 °C. Half-strength Hoagland solution (pH 5.6) was used to irrigate the seedlings. For Se treatment, pepper seedlings at the four true leaf stages were sprayed for 24 h with 0 or 100 ppm Na$_2$SeO$_4$ in 1/2 Hoagland solution.

### 2.2. Protein Extraction and Pancreatic Enzyme Hydrolysis

As described in our previous study [22], protein extraction was performed. In brief, liquid nitrogen was used to grind leaf tissue to powder. Four times the amount of powdered phenol extraction buffer (1% protease inhibitor, 1% phosphatase inhibitor, 10 mM dithiothreitol) was added and then ultrasonically lysed. The Tris equilibrium phenol was added

in equal volume and centrifuged for 10 min at 4 °C at 5500 g. Five times the volume of 0.1 M ammonium acetate/methanol was added to the supernatant to precipitate overnight. Finally, the precipitate was re-dissolved with 8 M urea.

A final concentration of 5 mM of dithiothreitol was added to the protein solution. Afterwards, iodoacetamide was added to achieve a final concentration of 11 mM and incubated at room temperature (25 °C) in darkness for 15 min. Finally, the urea concentration of the sample was diluted to below 2 M. Pancreatin was added in a mass ratio of 1:50 (pancreatin:protein) and hydrolyzed overnight at 37 °C. The mass ratio of trypsin to protein was 1:100 (trypsin:protein). The enzymatic hydrolysis was continued for 4 h.

### 2.3. TMT Labeling, HPLC Fractionation, and Phosphorylated Peptide Enrichment

The peptide segments that were hydrolyzed by trypsin underwent desalination using Strata X C18 (Phenomenex, Torrance, CA, USA) and were subsequently vacuum freeze-dried. The dissolved peptide segment was then labeled in accordance with the TMT kit operating instructions, after which the peptides were subjected to high pH reverse HPLC for grading. The chromatographic column employed was Thermo Betasil C18, measuring 5 μm in diameter, 10 mm in inner diameter, and 250 mm in length. The procedure involved the following steps: A peptide grading gradient of 8–32% acetonitrile at pH 9.0 was employed to separate 60 components over a duration of 60 min. Subsequently, the peptide was consolidated into 8 components, which were then subjected to vacuum freeze-drying for further processing.

The peptide segment was dissolved in an enriched buffer solution consisting of 50% acetonitrile and 6% trifluoroacetic acid. The resulting supernatant was then transferred to pre-washed IMAC material and placed on a rotating shaker for gentle shaking and incubation. Following incubation, the resin was washed three times using buffer solutions containing 50% acetonitrile and 6% trifluoroacetic acid, as well as 30% acetonitrile and 0.1% trifluoroacetic acid. Subsequently, 10% ammonia water was employed to elute the phosphate peptide, with the eluent being collected and subjected to vacuum freeze drying. Following the drying process, the elimination of salt was carried out in accordance with the guidelines provided in the C18 ZipTips instruction manual. Subsequently, the sample underwent vacuum freeze drying and was prepared for analysis using liquid chromatography–mass spectrometry.

### 2.4. Liquid Chromatography-Mass Spectrometry Analysis

The peptide segment was dissolved in a liquid chromatography mobile phase A, which consisted of a 0.1% ($v/v$) formic acid aqueous solution. Subsequently, it was separated utilizing the EASY-nLC 1000 ultra-high-performance liquid phase system. The mobile phase B consisted of an aqueous solution comprising 0.1% formic acid and 90% acetonitrile. The liquid phase gradient was set as follows: from 0 to 40 min, the concentration of B ranged from 4% to 22%; from 40 to 52 min, it ranged from 22% to 35%; from 52 to 56 min, it ranged from 35% to 80%; and from 56 to 60 min, it remained at 80%. Throughout the entire process, the flow rate was maintained at 450 nL/min. The peptide segments were separated by an ultra-high-performance liquid chromatography system and injected into an NSI ion source for ionization, followed by analysis by Q Activate™ Plus mass spectrometry. The ion source voltage was adjusted to 2.0 kV, while the peptide parent ions and their secondary fragments were subsequently detected and analyzed utilizing the high-resolution Orbitrap instrument. The primary mass spectrometry scanning range was configured to encompass 350–1800 $m/z$, with a scanning resolution of 70,000. The scanning range for secondary mass spectrometry was established with a fixed initial point of 100 $m/z$, while the Orbitrap scanning resolution was configured at 35,000. The data collection mode employed a Data Dependent Scanning (DDA) program, which identified and selected the top 10 peptide parent ions exhibiting the highest signal intensity subsequent to the initial level scanning. These selected ions were then subjected to fragmentation in the HCD collision pool, utilizing a fragmentation energy of 31%. Subsequently, secondary mass

spectrometry analysis was conducted on the fragmented ions. In order to improve the effective utilization of mass spectrometry, the automatic gain control (AGC) was set to $5 \times 10^4$, the signal threshold was set to 20,000 ions/s, the maximum injection time was set to 100 ms, and the dynamic exclusion time of tandem mass spectrometry scanning was set to 30 s to avoid repeated scanning of the parent ions.

### 2.5. Database Search and Bioinformatics Methods

The secondary mass spectrometry data were obtained through the utilization of Maxquant (v1.5.2.8). The retrieval parameter settings involved the utilization of the UniProt *Capsicum annuum* L. database, which consisted of 35,548 sequences. Additionally, a reverse library was incorporated to assess the false positive rate (FDR) resulting from chance matches. In order to mitigate the influence of contaminated proteins on the identification outcomes, the database had been supplemented with commonly encountered contamination libraries. The enzyme digestion method had been specified as Trypsin/P, while the number of allowable missing positions had been restricted to 2. Additionally, the minimum length of the peptide segment had been established as 7 amino acid residues. The maximum modification number of peptide was 5; the tolerance for mass error of the primary parent ion in the first and main searches was 20 ppm and 5 ppm, respectively, and the tolerance for mass error of the secondary fragment ion was 0.02 Da. The alkylation of cysteine was designated as a fixed modification, while the oxidation of methionine, deacylamination of asparagine and glutamine, acetylation of N-terminal protein, and phosphorylation of serine, threonine, and tyrosine were considered variable modifications. The quantitative approach employed was TMT-6plex, with a false discovery rate (FDR) of 1% for both protein identification and peptide-spectrum match (PSM) identification.

Gene ontology (GO) annotations were performed using the UniProt-GOA database (http://www.ebi.ac.uk/GOA/, accessed on 2 March 2023). On the basis of protein sequence alignment, InterPro 95.0 (http://www.ebi.ac.uk/interpro/, accessed on 8 March 2023) was used to annotate domain functional descriptions for the identified proteins. Protein pathways were annotated using the Kyoto Encyclopedia of Genes and Genomes (KEGG) database (http://geneontology.org/, accessed on 15 March 2023). WoLF PSORT (http://www.genscript.com/psort/wolf_psort.html, accessed on 16 March 2023) was used to predict subcellular localization. MoMo (motif-x algorithm) software (Version 3) (http://alumni.cs.ucr.edu/~mueen/OnlineMotif/index.html, accessed on 20 March 2023) was used to analyze the model of sequences constituted with amino acids in specific positions of modify-13-mers (6 amino acids upstream and downstream of the site) in all protein sequences. The selection of differential loci adhered to the subsequent criteria: a change threshold of 1.5 times and a *t*-test *p*-value less than 0.05.

### 3. Results

### 3.1. Analysis of Primary MS Data and Quantitative Phosphorylation Proteomics

Firstly, quality control for phosphorylation sequencing was conducted. The primary mass error of the vast majority of spectra was within 10 ppm, which conformed to the high-precision characteristics of orbital trap mass spectrometry (Figure 1A). Most peptide segments were distributed between 7 and 20 amino acids, following a general pattern based on trypsin enzymatic hydrolysis and HCD fragmentation (Figure 1B). Through mass spectrometry analysis, a total of 83,631 secondary spectra were obtained. The available effective spectrum number was 7672 following the search for protein theoretical data in the mass spectrometry secondary spectrum library, and the spectrum utilization rate was 9.2%. Through spectrum analysis, a total of 3872 peptide segments and 3468 phosphorylated peptide segments were identified. In all, 4434 phosphorylation sites on 2058 proteins were identified, of which 3749 sites on 1919 proteins had quantitative information (Figure 1C). The paired Pearson correlation coefficient between the six samples displayed good reproducibility among the three biological samples (Figure 1D).

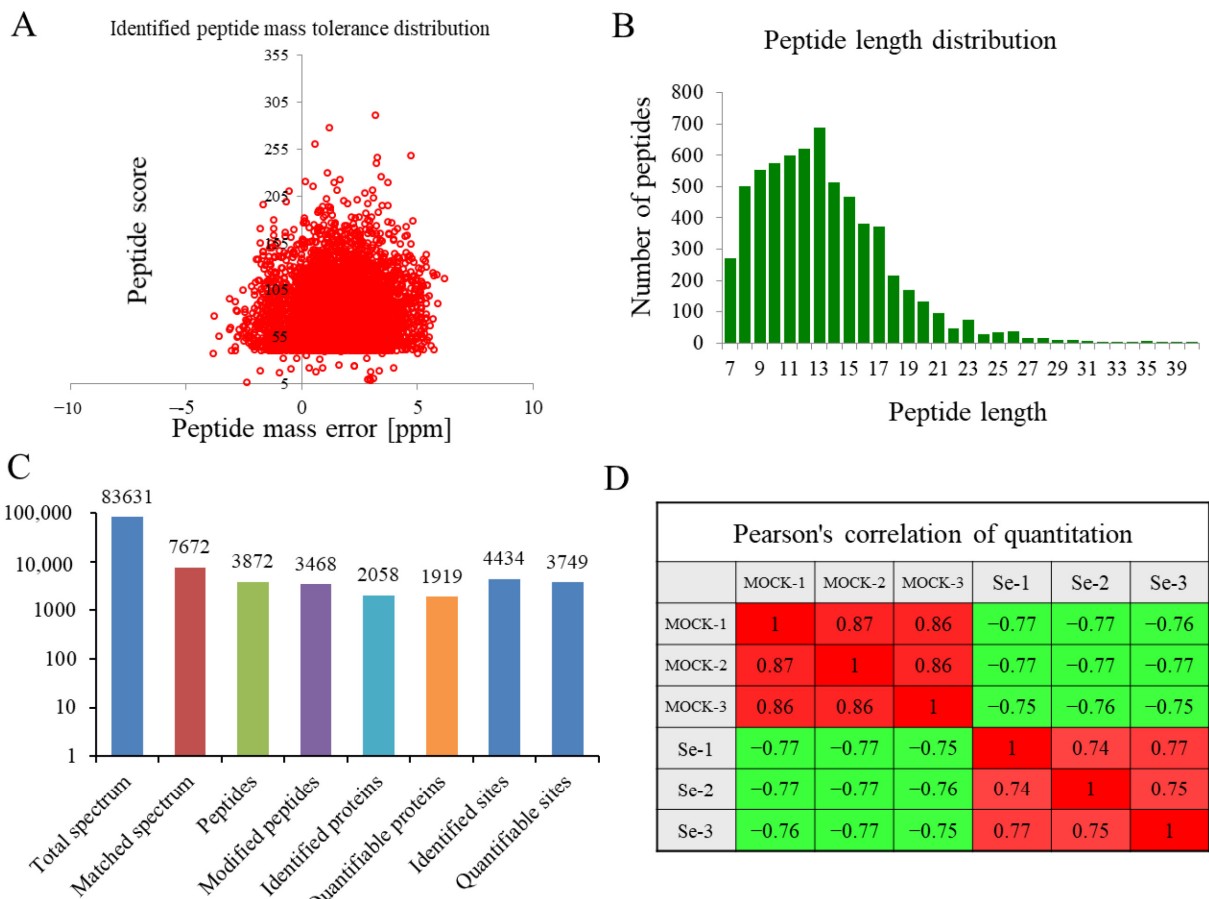

**Figure 1.** Quality control (QC) validation of mass spectrometer (MS) data. (**A**) Precision distribution of MS data. (**B**) Distribution of phosphorylated peptides by length. (**C**) Basic statistical figure of MS results. (**D**) Heatmap of Pearson's correlation coefficients (red shows positive correlation, and green shows no correlation).

*3.2. Annotation and Classification of All the Phosphorylated Proteins Identified in Peppers*

To provide a thorough understanding of the identified and quantified modified proteins in the data, detailed annotations on their functions and characteristics from the perspectives of GO, KEGG pathway, and subcellular structural localization are provided (Figure 2). According to GO terms, phosphorylated proteins are annotated (Figure 2A). In the biological process, the top three categories were 'metabolic process' (490 proteins), 'cellular process' (476 proteins), and 'single-organism process' (230 proteins). 'Binding' (879 proteins), 'catalytic activity' (536 proteins), and 'transporter activity' (63 proteins) were the top three terms in the molecular function category. 'Cell' (186 proteins), 'membrane' (141 proteins), and 'organelle' (99 proteins) contained the largest proteins in the cellular component category. There were four major categories based on KOG annotation of phosphorylated proteins (Figure 2B). For the 'information storage and processing' category, the main terms were 'RNA processing and modification' (88 proteins). The largest number of phosphorylated proteins were under the 'signal transduction mechanisms' term (262 proteins) in the 'cellular processes and signaling' category. A total of 56 phosphorylated proteins belonged to the 'carbohydrate transport and metabolism term in the 'metabolism' category. In total, 14 subcellular component categories were identified (Figure 2C), of which the number of proteins in nucleus, chloroplast, and cytoplasm was 834, 388, and 311, respectively. All identified proteins are presented in Table S1, including their protein IDs, subcellular localizations, GO, and KEGG.

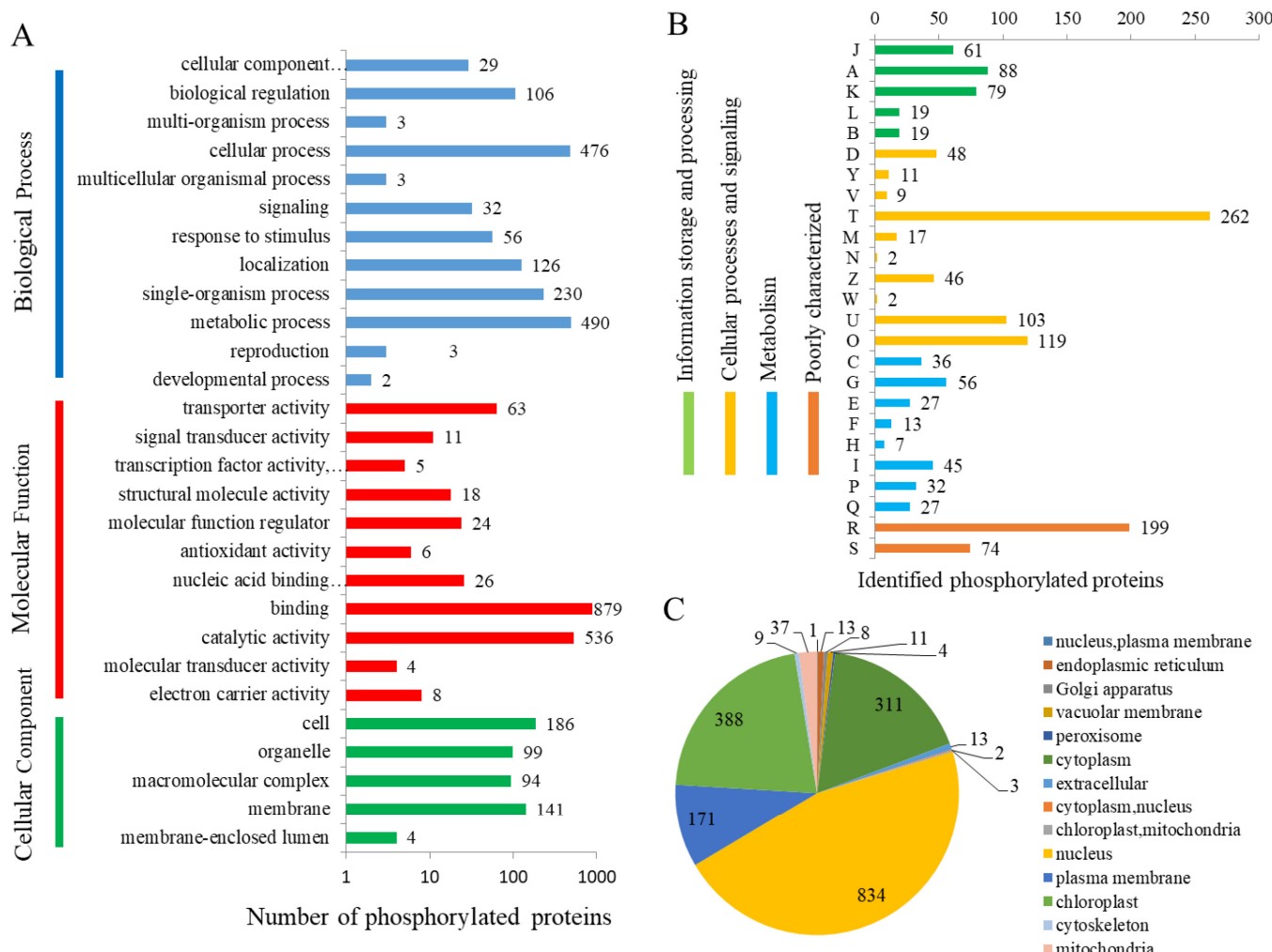

**Figure 2.** The annotation and classification of phosphorylated proteins. (**A**) GO terms analysis of phosphorylated proteins. (**B**) KOG analysis of phosphorylated proteins. (**C**) Subcellular locations of phosphorylated proteins.

### 3.3. Analysis of Phosphorylation Sites

It was found that 1080 phosphorylated proteins contained a single site, 455 phosphorylated proteins contained two sites, and 523 phosphorylated proteins contained three sites or more (Figure 3A). A total of 2837 of these residues was centered on a serine, 356 on a threonine, and 13 on a tyrosine (Figure 3B). A total of 33 over-represented motifs were found in the category of serine phosphorylation. The most frequent motifs were 'sP' (308 occurrences) and 'Gs' (182 occurrences), followed by 'Rxxs' (156 occurrences) (Figure 3C). In the category of threonine phosphorylation, 'tP' (101 occurrences) was the most frequent motif (Figure 3D).

### 3.4. Differentially Phosphorylated Proteins (DPPs) Responding to Selenium Treatment

The accumulation level of DPPs in peppers under selenium stress treatment is shown in the heat map (Figure 4A). Between treatment and control seedlings, 719 proteins (969 phosphorylated sites) were identified as DPPs, including 658 upregulated proteins (897 phosphorylated sites) and 61 downregulated proteins (72 phosphorylated sites) (Figure 4A,B). According to their subcellular localization, DPPs were classified into 13 subcellular components, including 330 nuclear-localized DPPs, 154 chloroplast-localized DPPs, and 122 cytoplasm-localized DPPs (Figure 4C). In addition, differentially phosphorylated transcription factors were also identified, including WRKY, bHLH, and NAC transcription factors (Table S2).

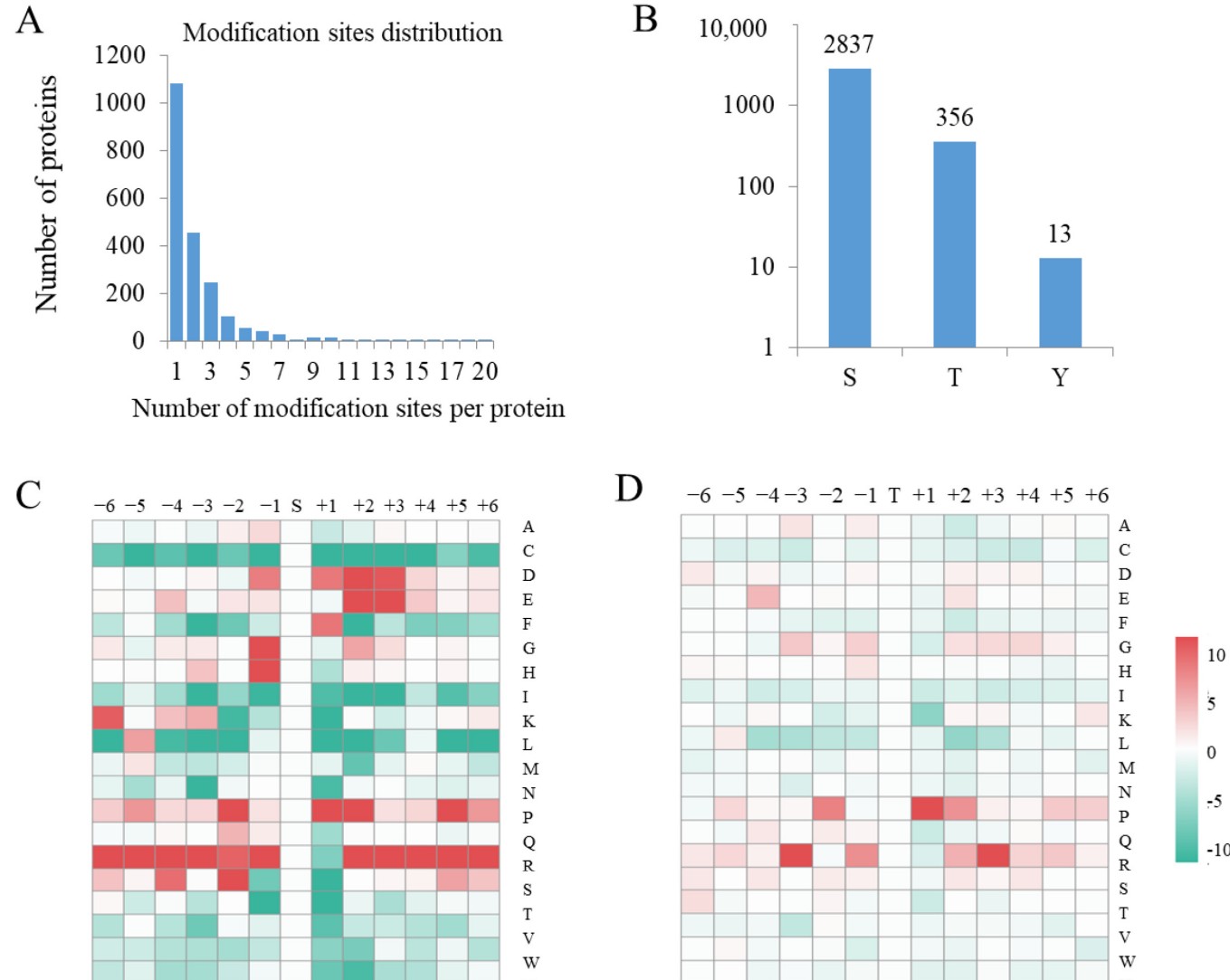

**Figure 3.** Analysis of phosphorylation sites. (**A**) The number of phosphorylated sites of all identified peptides. (**B**) The distribution of phosphosites in three amino acid residues (serine (S), threonine (T), and tyrosine (Y)). (**C**,**D**) A motif analysis of amino acids surrounding phosphorylated residues. The sequence logo shows 5 conserved phosphorylation motifs with S- and T-bases. The motif enrichment heatmap of all identifies phosphorylation modification sites upstream and downstream amino acids. Red represents a significant enrichment of this amino acid near the modification site, while green represents a significant reduction in this amino acid near the modification site.

*3.5. Enrichment Analysis of the DPPs*

　　To detect whether differentially modified proteins exhibit significant enrichment trends in some functional types, enrichment analysis of GO (Figure 5A) and KEGG (Figure 5B) pathways were performed. For upregulated proteins, eight GO terms were enriched in molecular function. 'Carbon–carbon lyase activity', 'tubulin binding' and 'MAP kinase activity' were top three most enriched terms. For downregulated proteins, four GO terms were enriched in molecular function. All DPPs were also classified by GO terms (Figure S1). A total of 24 PPS (30 phosphorylation sites) were grouped into 'response to stimulus' contained phosphoinositide phospholipase C, glutathione peroxidase, heat shock protein, and catalase (Table 1).

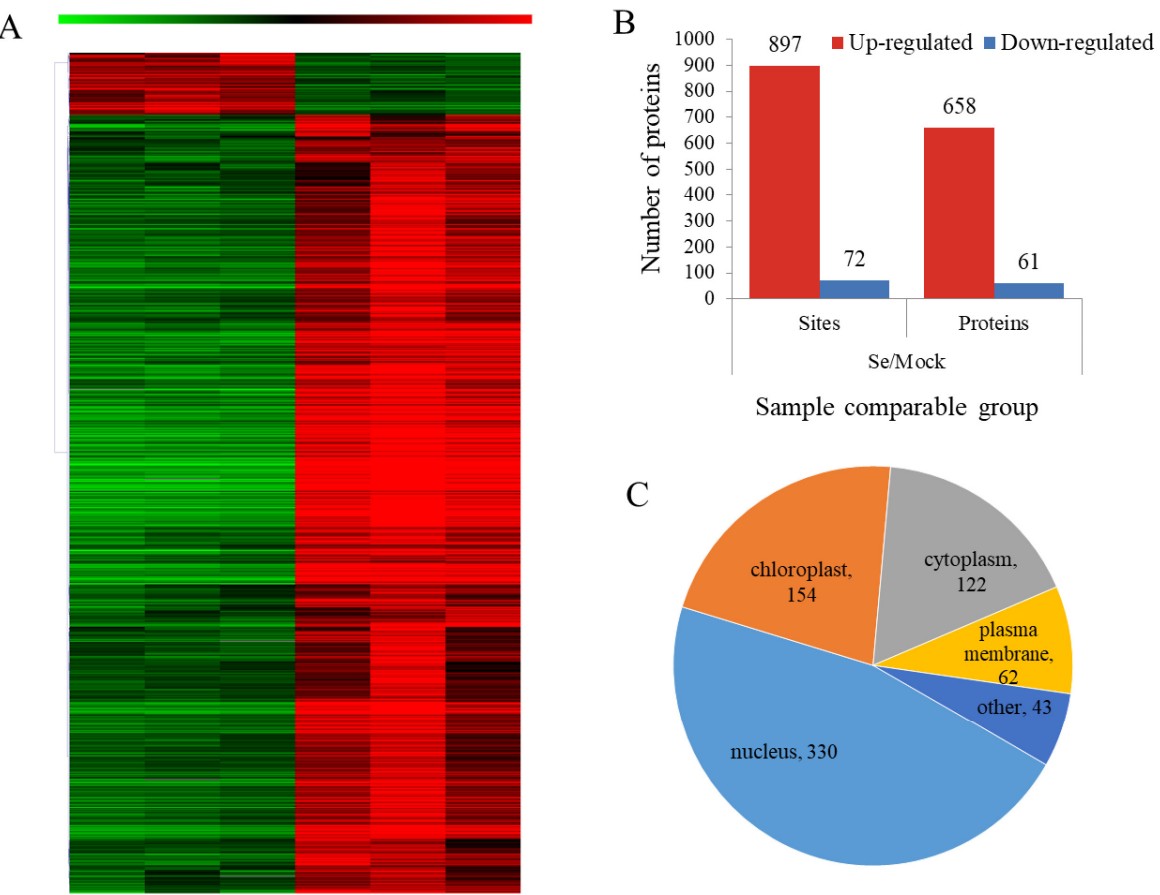

**Figure 4.** The impact of selenium stress on pepper phosphorylation proteome levels. (**A**) Heatmap of the DPPs in response to selenium stress. (**B**) The numbers of up- and downregulated phosphorylated proteins/sites in Se/mock (seedlings without Se treatment) comparison group. (**C**) Subcellular locations of the DPPs.

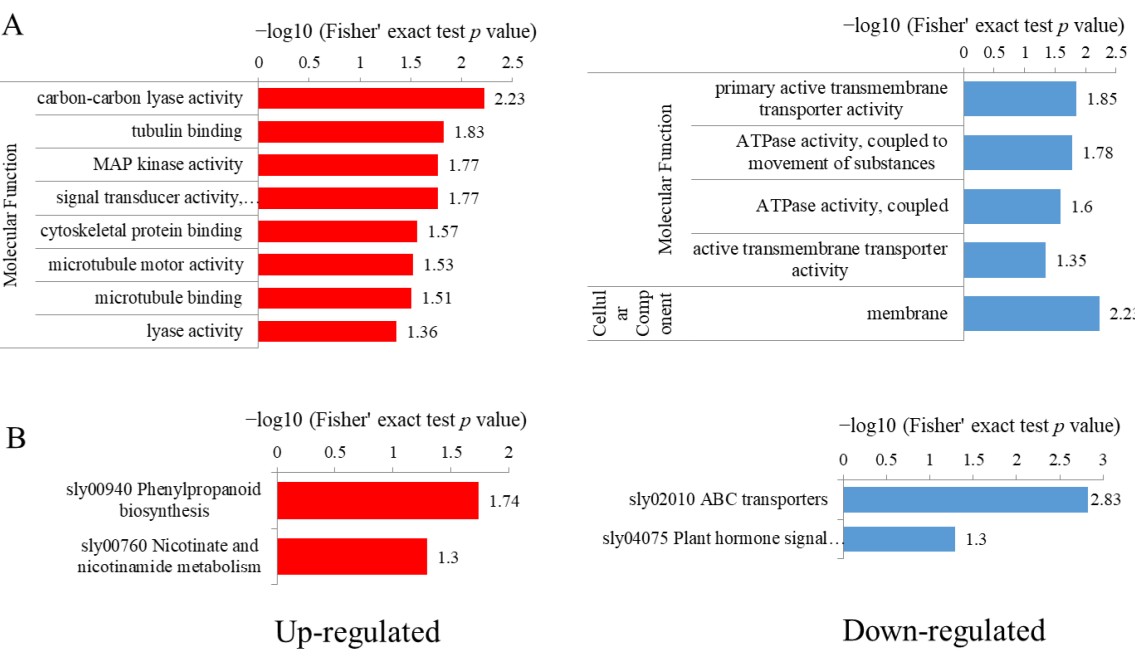

**Figure 5.** GO (**A**) and KEGG (**B**) enrichment analysis of the DPPs in pepper under selenium stress.

**Table 1.** Differentially expressed proteins related to 'response to stimulus' of GO term under selenium stress in pepper.

| Protein Accession | Position | Regulated Type | Amino Acid | Protein Description | Gene Name | Modified Sequence | Mock | Se |
|---|---|---|---|---|---|---|---|---|
| A0A2G2ZB33 | 17 | Up | S | Uncharacterized protein | T459_17272 | AIS(0.975)ELT(0.025)QGR | 0.49 | 1.51 |
| A0A2G2Z9L9 | 73 | Up | S | Uncharacterized protein | T459_16748 | SDMGS(0.002)LQNS(0.998)PR | 0.49 | 1.51 |
| A0A2G3AE16 | 153 | Up | T | Glutathione peroxidase | T459_00315 | YS(0.007)PT(0.117)T(0.759)S(0.117)PAS(0.001)MEK | 0.58 | 1.42 |
| A0A1U8ESJ2 | 50 | Up | S | Uncharacterized protein | T459_33300 | LSHFEMDHEGES(1)LK | 0.78 | 1.22 |
| A0A2G3A3R3 | 39 | Up | S | Vacuolar cation/proton exchanger 2 | T459_10985 | IDSLHYEAPHIVS(1)PR | 0.73 | 1.27 |
| A0A1U8FQK8 | 461 | Up | S | MLO-like protein | MLO | ALGNGS(1)PR | 0.77 | 1.23 |
| A0A1U8FX70 | 128 | Up | S | Uncharacterized protein | T459_05982 | S(0.003)S(0.003)GGIIGS(0.992)PPS(0.002)VENSSLK | 0.80 | 1.20 |
| A0A2G2YKX7 | 290 | Up | S | Phosphoinositide phospholipase C | T459_25500 | AWGAEIS(1)DLTQK | 0.67 | 1.33 |
| A0A2G2YQ26 | 925 | Down | S | Uncharacterized protein | T459_22633 | YYS(1)LPDISGR | 1.25 | 0.75 |
| A0A2G2Z5Z3 | 259 | Up | S | Heat shock protein 90-2 | T459_20949 | EVSNEWS(1)LVNK | 0.77 | 1.23 |
| A0A1U8FK60 | 514 | Up | S | Serine/threonine protein phosphatase 2A regulatory subunit | T459_04826 | AASNEPVLVS(1)PR | 0.65 | 1.35 |
| A0A2G2YJZ6 | 128 | Up | T | Plasma membrane-associated cation-binding protein 1 | T459_25171 | VS(0.154)T(0.846)FIVIPEEEK | 0.80 | 1.20 |
| A0A2G2YQ26 | 748 | Up | S | Uncharacterized protein | T459_22633 | VPEPLINS(0.135)NMY(0.042)S(0.823)PK | 0.76 | 1.24 |
| A0A1U8G2S9 | 98 | Up | S | Serine/threonine-protein phosphatase | T459_08345 | LRPAGEPPS(1)PR | 0.36 | 1.64 |
| A0A2G2YWS6 | 12 | Up | S | Guanine nucleotide-binding protein alpha-1 subunit | T459_21440 | HY(0.028)S(0.971)QADDEENAQTAEIER | 0.63 | 1.37 |
| A0A2G2ZUY0 | 25 | Up | S | Guanine nucleotide-binding protein subunit gamma 1 | T459_07889 | HRIS(1)AELKR | 0.78 | 1.22 |
| J9Q173 | 451 | Up | S | MLO-like protein | MLO2 | GT(0.077)S(0.923)PVHLLR | 0.76 | 1.24 |
| A0A2G2YAT8 | 434 | Up | S | Catalase | T459_31096 | Y(0.003)RS(0.997)WAPDR | 0.44 | 1.56 |
| A0A2G2YJP7 | 153 | Up | S | Uncharacterized protein | T459_25072 | NS(0.009)T(0.01)LT(0.238)T(0.744)PPIS(1)PK | 0.75 | 1.25 |
| A0A2G2YRS0 | 246 | Up | S | Phosphoinositide phospholipase C | T459_23224 | EVS(1)DLKAR | 0.75 | 1.25 |
| A0A1U8GJX1 | 99 | Up | S | Extra-large guanine nucleotide-binding protein 3 | T459_00959 | IAGVT(0.18)S(0.82)PPS(0.5)QS(0.5)PR | 0.37 | 1.63 |
| A0A2G2Z9L9 | 130 | Up | S | Uncharacterized protein | T459_16748 | DFS(1)FEKR | 0.33 | 1.67 |
| A0A1U8ESJ2 | 35 | Up | S | Uncharacterized protein | T459_33300 | HILNIS(0.998)PS(0.002)K | 0.64 | 1.36 |
| A0A2G3AE16 | 154 | Up | S | Glutathione peroxidase | T459_00315 | YSPTT(0.022)S(0.887)PAS(0.091)MEK | 0.59 | 1.41 |
| A0A2G2Z950 | 219 | Up | S | Heat shock protein 82 | T459_16580 | QIS(1)DDEDDEPKK | 0.56 | 1.44 |
| A0A2G2YJZ6 | 223 | Up | S | Plasma membrane-associated cation-binding protein 1 | T459_25171 | VEAAPAAAAAAAPAPS(1)KA | 0.65 | 1.35 |
| A0A2G2Y2E0 | 1051 | Up | S | ATP-dependent DNA helicase Q-like 4A | T459_32407 | GS(0.001)LT(0.014)S(0.012)GKQS(0.972)PPR | 0.78 | 1.22 |
| A0A2G2YYE8 | 9 | Down | S | Rho GTPase-activating protein 3 | T459_22041 | S(0.026)KS(0.962)YT(0.012)FGR | 1.31 | 0.69 |
| A0A2G2Z5Z3 | 424 | Up | S | Heat shock protein 90-2 | T459_20949 | LGIHEDS(1)QNR | 0.66 | 1.34 |
| A0A1U8GMP3 | 297 | Up | S | Uncharacterized protein | T459_12607 | EVSPEAVS(1)PIAMK | 0.54 | 1.46 |

According to KEGG enrichment analysis (Figure 5B), the upregulated DPPs were most strongly associated with the 'phenylpropanoid biosynthesis' and 'nicotinate and nicotinamide metabolism' pathways, while the downregulated DPPs were most strongly associated with 'ABC transporters' and 'plant hormone signal transduction'. DPPs involved in phenylpropanoid biosynthesis are further summarized (Figure 6). Three proteins related to flavonoid biosynthesis and *p*-Hydroxyphenyl lignin biosynthesis of phenylpropanoid biosynthesis were identified, including phenylalanine ammonia-lyase (PAL), 4-coumarate-CoA ligase (4CL), and cinnamyl alcohol dehydrogenase (CAD). There were three phosphorylated sites identified in these proteins (Figure 6A). The expression of these three proteins is shown in Figure 6B. They were all upregulated compared with the mock seedlings.

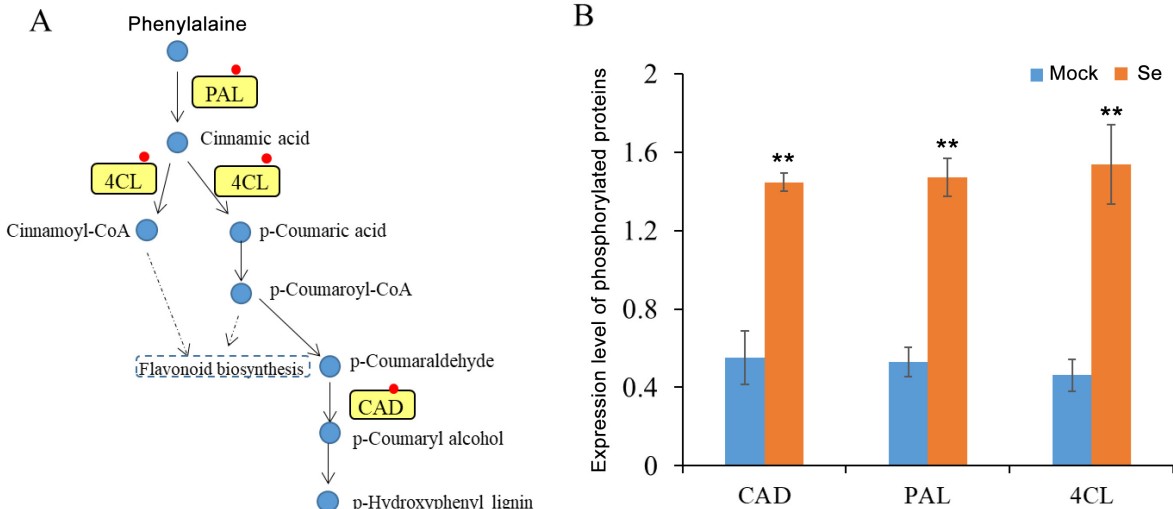

**Figure 6.** DPPs involved in phenylpropanoid biosynthesis. (**A**) Phenylpropanoid and flavonoid biosynthesis pathways in pepper. Yellow backgrounds indicate identified phosphorylated proteins. Red dots represent the number of phosphorylation sites in each phosphorylated protein. (**B**) The expression of DPPs involved in phenylpropanoid biosynthesis. The phosphorylation level represents means ± SD of three replicates. Mock: pepper seedlings without Se treatment. Asterisks indicate significant difference from the mock seedlings as determined by *t*-test at ** $p < 0.01$.

## 4. Discussion

Posttranslational modification of proteins is a common way to regulate protein function and endow it with regulatory specificity [28], including ubiquitination, sulfonylation, sulfation, glycosylation, and phosphorylation [29]. Among these, phosphorylation modification is the most common and widely studied post-translational modification, with one third of eukaryotic proteins considered to be phosphorylable [30]. To study pepper's response to selenium stress, a TMT-labeling quantitative phosphoproteomic method was conducted (Figure 1). A large number of phosphorylation sites of proteins that might be involved in the selenium stress response were identified (Figure 3). The numbers of phosphorylation sites and phosphorylated proteins were similar to those in pepper fruits [27]. The distribution of phosphorylation residues in pepper was found to be similar to other plants, with Ser phosphorylation sites accounting for over 80% and Tyr phosphorylation sites not exceeding 2% [31,32]. The Motif-X algorithm was used to identify 266 phosphorylation motifs in *Arabidopsis*, classified as 'proline directed', 'acidic', 'alkaline', and 'other'. This helped unravel the relationship between kinases, substrates, and kinase phosphorylation sites in *Arabidopsis* [33]. In chili peppers, many related motifs have also been identified. The [sP], [RxxsP], [GsP], [ssP], [Rxxs], [sDxE], and [tP] motifs are mainly involved in glucose metabolism, mRNA synthesis, and the MAPK (Mitogen activated protein kinase) pathway [34]. In this study, a total of 33 over-represented motifs were found in serine phosphorylation, including [sP], [Gs], and [Rxxs] (Figure 3C). In the category of threonine phosphorylation, 'tP' was the most frequent motif (Figure 3D). These results suggest that pepper plants may mediate multiple life processes in response to Se stress.

Through long-term evolution, plants have formed effective reactive oxygen species clearance mechanisms [35]. Catalase, which can participate in the metabolism of reactive oxygen species and prevent cell peroxidation, is related to plant stress response [36]. In animals, PKC δ and the kinase complex c-Abl/Arg phosphorylates catalase, thereby regulating the polymerization state and stability of catalase [37]. In the present study, the results also showed an increase in phosphorylated catalase content after selenium treatment (Table 1). Hsp is a kind of molecular chaperone protein produced by plants under the stress of biotic or abiotic factors [3,38]. Three HSPs were identified among the Se-stress-induced DPPs in pepper (Table 1), indicating the molecular interaction between Se

stress and heat shock response. Phosphoinositide specific phospholipase C is one of the key enzymes in the inositol phospholipid signaling system and plays a part in plant adaptation to abiotic stress [39]. In the signal transduction of plant cells, phosphoinositide specific phospholipase C and its hydrolysates play an important mediating role [40]. Our results indicated that phosphoinositide specific phospholipase C were induced by selenium stress (Table 1), implying that selenium signaling regulates selenium stress response through a signaling pathway mediated by phosphoinositide specific phospholipase C.

The general assessment of the regulation of metabolites and plant hormone under stress has been widely discussed [41]. Flavonoids are one of the most abundant secondary metabolites in plants [42]. Plants respond to different stresses by regulating the synthesis and accumulation of flavonoids, thus forming a self-protection mechanism [36]. PAL, 4CL, and CAD are key enzymes in flavonoid synthesis, and there were three phosphorylated sites identified in these proteins (Figure 6A). These results indicated that selenium stress affected the phosphorylation of synthetic proteins related to phenylpropanoid biosynthesis and flavonoid synthesis, thereby regulating the accumulation of flavonoids (Figure 6). ABCB, as a transporter, cooperates with AUX1/LAX (AUXIN1/LIKEAUXIN) and PIN (PIN-FORMED) to regulate the polar transport of plant auxin [43]. The change of auxin transport vector expression changes the accumulation of auxin, thus affecting the resistance of plants to stress [43,44]. The results in this paper showed that selenium affected the phosphorylation of two ABCB proteins (Figure 5), which may have affected the distribution of auxin.

Phosphorylation can modify some transcription factors and control their activity by changing their conformation, thereby affecting the expression of downstream genes. WRKY family genes can regulate multiple responses through a complex genetic network [45]. Some WRKY genes can improve plant salt and drought tolerance by regulating stomatal openings and ROS levels [46,47]. The phosphorylation of bHLH Ia subfamily affects stomatal differentiation, the phosphorylation state of bHLH VIIa subfamily affects plant photoresponse, and the phosphorylation of bHLH IIIa subfamily affects plant iron acquisition [48]. Overexpression of *OsMAPK3* leads to phosphorylation and enhanced cold tolerance of bHLH002 [49]. Replacing serine/threonine at different positions with alanine resulted in phosphorylation death mutants (bHLH002T404A, bHLH002T406A, bHLH002S407A, bHLH002T412A, and bHLH002S433A). At this time, the cold tolerance function of the mutant was lost, indicating that OsMAPK3-mediated OsbHLH002 phosphorylation enhanced its function in cold signal transduction. As an important transcription regulatory factor, NAC transcription factors not only participate in regulating plant life activities, but also are in a complex regulatory network. In specific situations, they can be regulated at the transcription level, post transcription level, and translation and post translation levels [50]. In this study, differentially phosphorylated transcription factors including WRKY, bHLH, and NAC were also identified (Table S2), suggesting these transcription factors may be phosphorylated to regulate Se stress response.

## 5. Conclusions

In summary, to understand the role of pepper protein phosphorylation in signal transduction under selenium stress, a comprehensive phosphorylation proteomic analysis was performed by TMT. A total of 4434 phosphorylation sites were identified on 2058 proteins. Compared with the mock seedlings, 719 proteins (969 phosphorylated sites) were identified as DPPs, including 658 upregulated proteins (897 phosphorylated sites) and 61 downregulated proteins (72 phosphorylated sites) under Se stress. DPPs involved in phenylpropanoid biosynthesis, including PAL, 4CL, and CAD, were further identified. Moreover, a large number of transcription factors (WRKY, bHLH, and NAC) may be phosphorylated to regulate Se stress response.

**Supplementary Materials:** The following supporting information can be downloaded at: https://www.mdpi.com/article/10.3390/horticulturae9080935/s1, Table S1: The detail annotation information of all the phosphorylated proteins; Table S2: Differentially phosphorylated transcription factors identified under selenium stress in pepper; Figure S1: Statistical distribution chart of proteins corresponding to differentially expressed modification sites under each GO category.

**Author Contributions:** Conceptualization, J.W. (Jiahua Wu) and Y.Z.; methodology, J.W. (Jiahua Wu); software, S.G.; validation, J.W. (Jing Wang), J.H. and X.L.; writing—original draft preparation, Y.Z.; writing—review and editing, Y.Z.; project administration, S.G.; funding acquisition, Y.Z. All authors have read and agreed to the published version of the manuscript.

**Funding:** This research was funded by the Scientific Research Foundation of Zhejiang A&F University (2020FR057).

**Data Availability Statement:** Not applicable.

**Conflicts of Interest:** The authors declare no conflict of interest.

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
