# Peer review of "The Phosphoproteomic Response of Pepper (Capsicum annuum L.) Seedlings to Selenium Stress"

_horticulturae, doi:10.3390/horticulturae9080935_

Round 1
Reviewer 1 Report
This work has well investigated the proteomic phosphorylation of pepper induced by selenium stress. However, stress-induced (de-)phosphorylation of protein-based biological molecules will certainly modify their function on related signaling pathway as a part of stress responses. In the introduction and discussion of this work, only a very generalized demonstration was given on the effects of phosphorylation on the functions of certain protein transporter/enzymes/signaling pathway, which is very imbalanced compared to abundant data presented in this work. Namely, the data were poorly explained considering plenty of DPPs induced by Se stress detected throughout the work and significantly jeopardized the value of the results and quality of the paper. More background, explains, and discussion should be given to improve the significance indicated by these data for the sake of readerships.
Additionally, some minor issues mentioned below have to be paid attention:
Line 56 Please confirm any metabolism pathway “in plasmid”?
Line 76 Please give the full name of ‘TMT’.
Line 95. Please avoid the use of Arabic number as the first word of the sentence.
Line 107-108, “…with dimensions of 5 μm in diameter, 10 mm in diameter, and 250 mm in length.” Herein, which one is which diameter?
Line 174, ‘analyze’
In Figure 6, ‘phenylalaine’
Line 265-269, please pay attention to the ‘capitalized’ letters.
Line 272, where is the demonstration of figure 6B??
Line 302-303, 310-312…, the results in this study need to be referred to certain table/figure/supplementary materials. This also applies to the rest content of the discussion when discussing about current results/conclusions.
Some lauguage error may lead to significant misunderstanding of the content. Authors should recheck the intergity of sentences used carefully.
Reviewer 2 Report
1. What is the main question addressed by the research?
In addition to the previously mentioned, I can add that the authors also aimed to present the molecular mechanism of selenium identification in plants with the aim of promoting selenium-rich foods and providing a theoretical basis for the absorption of selenium from plants by the human body . The authors explore the role of protein phosphorylation in selenium signal transduction in Pepper (Capsicum annuum L.) plants.
2. Do you consider the topic original or relevant in the field? Does it address a specific gap in the field?
The study of stress caused by excess selenium in plants has been little studied. There are some works in this regard on wheat (Luo, F., Zhu, D., Sun, H., Zou, R., Duan, W., Liu, J. and Yan, Y., 2023. Wheat Selenium-binding protein TaSBP-A enhances cadmium tolerance by decreasing free Cd2+ and alleviating the oxidative damage and photosynthesis impairment. Frontiers in Plant Science, 14),
to green algae (Zhang, B., Duan, G., Fang, Y., Deng, X., Yin, Y. and Huang, K., 2021. Selenium (Ⅳ) alleviates chromium (Ⅵ)-induced toxicity in the green alga Chlamydomonas reinhardtii. Environmental Pollution, 272)
or, in general, to plants (Bolea-Fernandez, E., Balcaen, L., Resano, M. and Vanhaecke, F., 2015. Interference-free determination of ultra-trace concentrations of arsenic and selenium using methyl fluoride as a reaction gas in ICP–MS/MS. Analytical and bioanalytical chemistry, 407),
especially related to the toxicity of other chemical elements such as arsenic (Chitta, K.R., Landero Figueroa, J.A., Caruso, J.A. and Merino, E.J., 2013. Selenium mediated arsenic toxicity modifies cytotoxicity, reactive oxygen species and phosphorylated proteins. Metallomics, 5(6), pp.673-685),
but there are no specific studies on absorption and food availability of selenium in pepper. Therefore, the peer-reviewed paper is relevant in this field and complements the specific scientific information.
3. What does it add to the subject area compared with other published material?
The present work brings new information regarding the response of plants to the stress caused by some metals through the quantitative proteomic determination of phosphorylation and TMT labeling – modern methods applied to a species present in food, especially as a spice (pepper). The authors performed a systematic bioinformatics analysis, including protein annotation, functional classification, subcellular localization, and cluster analysis based on the functional enrichment of proteins containing quantitative information sites.
4. What specific improvements should the authors consider regarding the methodology? What further controls should be considered?
I consider that the methodology is current and correct and I do not think that anything more is needed for this work.
5. Are the conclusions consistent with the evidence and arguments presented and do they address the main question posed?
The paper does not have a separate Conclusions chapter. It would be good if the authors add this chapter and clearly present some conclusions. They presented in chapter "4. Discussion" some conclusions, but they should be moved to the separate dedicated chapter. It should also clearly respond to the proposal in the title of the paper.
6. Are the references appropriate?
Yes, the papers cited refer to the research done and are recent. However, in the text of the previous review I recommended another work to be studied on this topic.
The bibliography could also be completed, for example with the paper: Liu, Z., Lv, J., Liu, Y., Wang, J., Zhang, Z., Chen, W., Song, J., Yang, B., Tan, F., Zou, X. and Ou, L., 2020. Comprehensive phosphoproteomic analysis of pepper fruit development provides insight into plant signaling transduction. International Journal of Molecular Sciences, 21(6).
7. Please include any additional comments on the tables and figures.
I have no additional comments regarding the figures and tables presented, including the additional files.
Reviewer 3 Report
The manuscript is important and has potential for publication. The authors analyzed phosphorylated proteins that are related to the stress caused by selenium in pepper seedlings. Some suggestions and corrections are presented below:
- In the abstract:
I encourage to author mainly focus the results and research insight in the abstract. Author should also consider that the abstract should be short and concise.
The keywords have to be different from the title. To replace.
_ In the Introduction: Write hypotheses about the work before the objectives. The realization of few studies with this theme does not justify research on.
- In Material and Methods:
Describe the study site in more detail.
- In Results and Discussion
Remove bold from all Figures.
A topic on Conclusion was missing, showing the main results and conclusion of the study.
Reviewer 4 Report
Please check in document

Round 2
Reviewer 1 Report
Line 307, is '5 μm' or '10mm' the inner diameter?
Figure 6, it is apparent that the values of individual DPPs between Mock and Se groups are significantly different with each other, but it is the reason to leave the place of sitatistical labels empty.
The language has been improved.
Author Response
Point 1: Line 307, is '5 μm' or '10 mm' the inner diameter?
Response 1:. The inner diameter is 10 mm.
Point 2: Figure 6, it is apparent that the values of individual DPPs between Mock and Se groups are significantly different with each other, but it is the reason to leave the place of sitatistical labels empty.
Response 2: We have added the sitatistical labels in Figure 6.
Reviewer 3 Report
The authors accepted the suggestions.
Author Response
Thank you very much for your comments.
Reviewer 4 Report
The comments were attended by authors
Author Response
Thank you very much for your comments.